# Artificial Light at Night Influences Clock-Gene Expression, Activity, and Fecundity in the Mosquito *Culex pipiens* **f.** *molestus*

**Ann-Christin Honnen** [1,2,3,*] , **Janina L. Kypke** [1,4] , **Franz Hölker** [1,5] and
**Michael T. Monaghan** [1,5,6,*]

1   Leibniz-Institute of Freshwater Ecology and Inland Fisheries (IGB), Müggelseedamm 301/310, 12587 Berlin, Germany; jkypke@hotmail.com (J.L.K.); hoelker@igb-berlin.de (F.H.)
2   Swiss Tropical and Public Health Institute, Socinstrasse 57, P.O. Box, CH-4002 Basel, Switzerland
3   University of Basel, Petersplatz 1, P.O. Box, CH-4001 Basel, Switzerland
4   Natural History Museum of Denmark, Zoological Museum, Universitetsparken 15, DK-2100 Copenhagen, Denmark
5   Institut für Biologie, Freie Universität Berlin, Königin-Luise-Str. 1-3, 14195 Berlin, Germany
6   Berlin Center for Genomics in Biodiversity Research, Königin-Luise-Str. 6-8, 14195 Berlin, Germany
*   Correspondence: a.honnen@swisstph.ch (A.-C.H.); monaghan@igb-berlin.de (M.T.M.);
    Tel.: +41-61-284-8821 (A.-C.H.); +49-30-6418-1684 (M.T.M.)

**Abstract:** Light is an important environmental cue, and exposure to artificial light at night (ALAN) may disrupt organismal physiology and behavior. We investigated whether ALAN led to changes in clock-gene expression, diel activity patterns, and fecundity in laboratory populations of the mosquito *Culex pipiens* f. *molestus* (Diptera, Culicidae), a species that occurs in urban areas and is thus regularly exposed to ALAN. Populations were kept under 16 h:8 h light:dark cycles or were subjected to an additional 3.5 h of light (100–300 lx) in the evenings. ALAN induced significant changes in expression in all genes studied, either alone (*period*) or as an interaction with time (*timeless*, *cryptochrome2*, *Clock*, *cycle*). Changes were sex-specific: *period* was down-regulated in both sexes, *cycle* was up-regulated in females, and *Clock* was down-regulated in males. ALAN-exposed mosquitoes were less active during the extra-light phase, but exposed females were more active later in the night. ALAN-exposed females also produced smaller and fewer eggs. Our findings indicate a sex-specific impact of ALAN on the physiology and behavior of *Culex pipiens* f. *molestus* and that changes in clock-gene expression, activity, and fecundity may be linked.

**Keywords:** Circadian clock; ALAN; Behavior; LED; Egg production; *period*; *Clock*; *cycle*; *cryptochrome2*; *timeless*

## 1. Introduction

Artificial light at night (ALAN) is a prominent feature of most urban and semi-urban areas. The urbanization of the human population, i.e., the number of people living in urban areas, is projected to increase from the current 4 billion to 6.3 billion people by 2050 [1]. This is accompanied by a yearly increase of artificially lit surface of Earth at night of 2–6%, both in radiance and extent [2,3]. In addition, there is an increase of regularly illuminated urban underground infrastructures such as metro systems. While ALAN has been found to affect the physiology and behavior of individual organisms, the extent to which this affects fitness remains poorly understood [4–7]. Studies of adult insects have focused on the attraction to artificial light [8–11], the loss of orientation and exhaustion [10–12], or dietary changes [13]. Less studied are the effects ALAN may have on diel rhythms of exposed organisms.

Light acts as a prominent environmental cue to set and synchronize circadian timekeeping with the ambient light cycles. This synchronization in turn regulates entire metabolic pathways. Disruption of the circadian clock can therefore lead to modifications in the timing of key behaviors such as foraging and mating [12,14,15], which may ultimately lead to altered fitness due to changes in energy uptake or changes in lifetime mating success [16,17].

Many mosquito species (Diptera, Culicidae) thrive in areas of human habitation [18] and are consequently exposed to ALAN. Understanding the effects of ALAN on physiology, behavior and fecundity of mosquitoes may thus have important implications for mosquito populations, including their function in natural ecosystems [19,20] and disease transmission dynamics [21]. In Europe, the widespread species *Culex pipiens* commonly occurs in urban and suburban areas [22,23] and is comprised of two forms or "ecotypes": *Cx. pipiens* f. *molestus* (hereafter molestus) and *Cx. pipiens* f. *pipiens* (hereafter pipiens) [24]. The molestus ecotype occurs in North America, Southern and Northern Europe. It was long believed to be restricted to underground habitats in its northern distribution range, e.g., New York and London [25–27]; however, recent studies have found that both ecotypes occur sympatrically above-ground [28]. *Culex pipiens* f. *molestus* readily mates in confined spaces and does not require a blood meal to produce eggs [24,26]. These particular features make it well-suited for controlled laboratory experiments. Although pipiens mates primarily in open spaces, it requires a blood meal before oviposition, and undergoes winter diapause [24]; there is demonstrated phenotypic plasticity in mating, as they are able to breed in the laboratory without swarm formation [23,29]. The two forms are not easily distinguished and even genetic diagnostic tests fail to separate them in some parts of their range [30]. *Culex pipiens* activity is typically crepuscular-nocturnal, with the daily peak of host-seeking activity having been observed to shift from night in summer to evening in autumn [31]. This is likely to expose urban populations of *Cx. pipiens* to ALAN because this coincides with the timing of their natural flying behavior [31,32]. In many rural and urban areas, there is a temporal dividing line around midnight, after which full illumination ends as some lights are extinguished [33]. Effectively, ALAN alters the photoperiod, providing less time for this nocturnal species to show behavior associated with the active phase of their circadian cycle.

The genetic basis of the mosquito circadian clock (Figure 1) includes the genes *Clock, cycle, period, timeless, cryptochrome1* and *cryptochrome2* [34–38]. These genes and their products comprise a central feedback-loop that rhythmically regulates transcription and repression of its components [39,40]. The circadian clock of mosquitoes differs from the well-studied *Drosophila* clock by the possession of another cryptochrome: *cryptochrome2*. This closely resembles the hypothesized clock model of the Monarch butterfly (*Dannaus plexippus*), which also possesses two cryptochromes. The possession of two cryptochromes is assumed to be the ancestral state of the clock in insects [41] and this model is likely to be true for other insects with two cryptochromes [42]. Therefore, it is assumed that the mosquito clock functions similar to that of the monarch butterfly (Figure 1). In the monarch butterfly, and likely also in mosquitoes, the proteins CLOCK and CYCLE form a dimer in the nucleus that binds to the E-Box promoting the expression of *period, timeless* and *cryptochrome2*). The proteins migrate into the cytoplasm where they form a complex. It remains unknown whether TIMELESS is part of this complex, stabilizing it, or whether CRYPTOCHROME2 interacts with PERIOD and TIMELESS in vivo [43]. Environmental light cues are received by the blue-light receptor CRYPTOCHROME1, leading to its activation [42]. This constitutes the pathway by which environmental light is perceived and affects the functioning of the circadian clock by entraining it to ambient light conditions. The activated CRYPTOCHROME1 then degrades TIMELESS while PERIOD is phosphorylated during the course of the night [44]. It is thought that PERIOD assists CRYPTOCHROME2 in migrating back into the nucleus where it represses the CLOCK-CYCLE-dimer, thereby repressing the transcription of *period, timeless* and *cryptochrome2* genes [43,44].

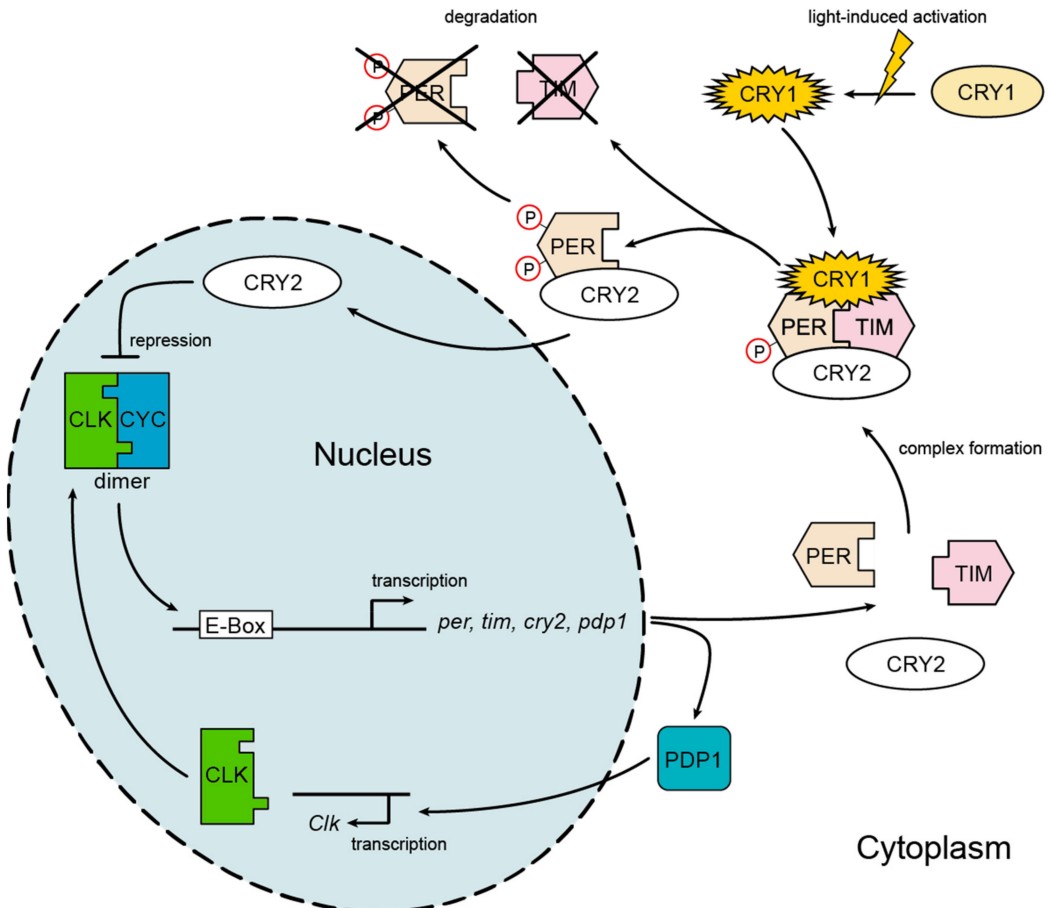

**Figure 1.** Hypothetical model of the central circadian clock feedback loop in Monarch butterflies (*Dannaus plexippus*), after Kyriacou [42]. The possession of two cryptochromes is assumed to be the ancestral state of the insect circadian clock [41] and this model is likely to be true for other insects with two cryptochromes [43], such as *Cx. pipiens*. Gene loci are indicated with lowercase italics and proteins are indicated by block capital letters, where *per* = *period*, *cry1* = *cryptochrome1*, *cry2* = *cryptochrome2*, *tim* = *timeless, Clk* = *clock, cyc* = *cycle, pdp-1* = *PAR-domain protein 1*, proteins are depicted in capital letters, genes are depicted in italics. Expression of the two cryptochromes and cycle is regulated in different, interlocked feedback loops.

Artificial light at night presents an alteration of the light regime that differs from what has been investigated in most studied organisms. A number of studies have identified important links between light and activity in mosquitoes [31,45,46], but we are not aware of any that examined how ALAN may affect physiology and activity. Because of the link of the circadian clock to downstream processes, it is reasonable to assume that light-mediated changes of the rhythm may lead to changes in activity patterns, which could then affect host-search, feeding, and mating. Altered feeding behavior could in turn influence the nutritional state of the individuals in several ways. Diurnal organisms may be able to use extra light to increase the time used for feeding, whereas nocturnal species may avoid light, thereby reducing available time for foraging.

Evidence of the effects of ALAN on fitness is scarce. ALAN is able to alter the timing of reproductive physiology and lay date in songbirds [47,48] and to suppress different hormones along the reproductive axis in common freshwater fish species [5]. In insects, ALAN was found to reduce sex pheromone production and mating in moths [49,50]. We are aware of only one study investigating how extra light in otherwise dark phases affects fecundity in Diptera, which reports a decrease in the number of eggs produced in *Drosophila melanogaster* [51]. A reduction in fecundity could provide

evidence that ALAN may affect population growth, which could further have an effect on disease epidemiology in vector species.

We used three laboratory experiments to test whether (1) clock gene expression, (2) behavior, and (3) fecundity in *Cx. pipiens* f. *molestus* were altered by 3.5 h of additional ALAN (100–300 lux) during the otherwise dark phase of the diurnal cycle. The aim was to mimic a lighting scheme typically found in urban environments whereby nocturnal individuals experience an extended dusk which could disrupt the normal cue for the onset of daily activities, e.g., foraging, during the night [31]. We expect that ALAN could affect all three tested responses, by providing brighter than natural light at dusk. In the first experiment, changes in expression were measured for five genes of the circadian clock relative to a constitutive gene over 10 time points from late afternoon until after midnight. In the second experiment, activity was continuously measured in light:dark cycles followed by measurements in constant darkness. Finally, population fecundity was assessed as mean egg size, numbers of eggs produced, and numbers of egg rafts.

## 2. Materials and Methods

### 2.1. Laboratory Colony

A laboratory colony of *Culex pipiens* f. *molestus* was started from eggs in April 2012. These eggs originated from an existing colony at the Bernhard-Nocht-Institute for Tropical Medicine in Hamburg, Germany, which was begun in July 2011 from larvae collected above-ground from a cemetery near Karlsruhe, Germany. The mosquitoes were reared in a climate-controlled chamber at 26 ± 1 °C with a relative humidity of ca. 60–90%. Adults were kept in mesh cages (60 × 30 × 30 cm) and fed with ≈10% sucrose solution offered on cotton pads *ad libitum*. Males and females were kept together to allow mating, and females were not blood-fed. Petri dishes were provided for oviposition and egg rafts were removed from the Petri dishes daily and placed in open trays (16 × 29 cm) filled with water. After hatching, larvae were fed with ground algal flakes (JBL Spirulina, JBL, Neuhofen, Germany). Pupae were transferred to water-filled beakers (50 mL) in the cages and allowed to emerge. Oviposition, emergence, and larval containers were filled with lake water that was collected weekly from nearby Lake Müggelsee. We used lake water because the copper content of the laboratory water source proved to be lethal to larval mosquitoes. Because lake water undergoes temporal changes in concentration of organic matter (detritus, plankton) and nutritional quality, we filtered (10 μm) and autoclaved the water before use.

The mosquitoes spent their entire life cycle in a climate-controlled chamber in which two independent light regimes were established. Males and females were kept in cages separated by age to ensure that the mosquitoes sampled for further analyses were approximately the same age. Due to space limitations mosquitoes were not separated by gonotrophic state. The first light regime, i.e., "control", had a light:dark (L:D) cycle of 16 h:8 h (Figure 2) designed to mimic early summer daylight hours at the study latitude (52°26′51.986″N). We chose relatively low experimental daylight conditions (300–800 lx) because the mosquito is well known for being commonly found in dim light structures such as bushes or even sewers during the day. Typical brightness values during the day range from a maximum of about 100,000 lx on clear days to 100–2000 lx on overcast days [52]. The experimental light regime ("extra-light" hereafter) had the same duration of 24 h L:D cycle, but with an extended 3.5 h period of ALAN (100–300 lux), effectively 15.5 h:3.5 h:5 h (L:dim:D), during the early part of the night (Figure 2). The mosquitos in the control experienced 0.0 lx. Typical brightness values during the night range from a maximum illuminance of 0.3 lx at a full-moon night [53], which decreases to about 0.001 lx at a moonless clear night [52] and even further for cloudy conditions [54]. This 3.5 h period began when light had decreased to 1% of the mid-day level, at which point light returned to 300 lux (Figure 2A) or 100 lux (activity experiment, Figure 2B) (transition time 10 s) and remained constant for 3.5 h in the "extra light" treatment. Light was produced using light-emitting diode (LED) illuminants (LED flex SMD, 24VDC, 24 W, 1 A, 60 LEDs/m, 500 cm, cool-white single chip, Barthelme

GmbH & Co. KG, Nuremberg, Germany) arranged in seven strips of 48 LEDs each. These were attached to a wooden board (88 × 34 cm) and suspended horizontally over the cages. Light levels were controlled with custom software based on the LabView runtime environment (National Instruments v 8.5.1, cf. Figure 2A. We specified voltage at 15 (control) or 19 (extra-light) time points to which the software fit a hermite spline curve. The result was a smooth change of light intensity over each 24-h interval (Figure 2A, Supplementary Materials Table S1). Light intensity was measured 75 cm below the illuminant board inside each cage on the bottom using a light meter (ILT1700, International Light Technologies, Peabody, MA, USA). The spectral composition of the LED used for the study is given in Supplementary Materials Figure S1.

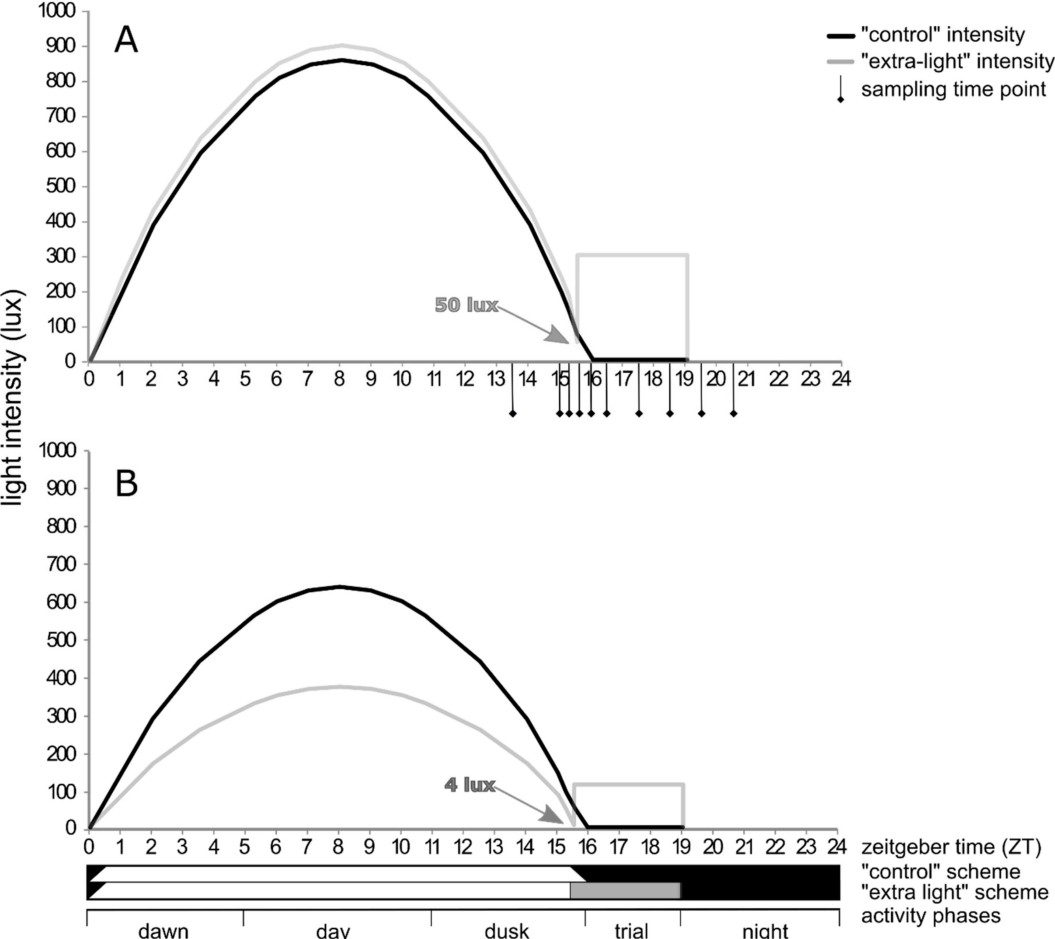

**Figure 2.** Light regime for the gene expression and fecundity experiments (**A**) and the activity experiment (**B**). Light source: cool-white LEDs; light temperature: 4000 K; spectral distribution shown in Figure S1. The upper x-axis indicates time in zeitgeber time (ZT) in hours, with ZT0 = lights on and ZT16 = lights off. Constant added light 300 lux (A, comparable to underground lighting [55]) or 100 lux (B, comparable to strong street lights [52]) in the extra-light treatment started at ZT 15.5 (=19:30 h) and ended at ZT 19 (=23:00). Markers on the x-axis indicate sampling times for the gene expression experiment (**A**): ZT 13.5 (=17:30), ZT 15 (=19:00), ZT 15.3 (=19:20), ZT 15.7 (=19:40), ZT 16 (=20:00), ZT 16.5 (=20:30), ZT 17.5 (=21:30), ZT 18.5 (=22:30), ZT 19.5 (=23:30), ZT 20.5 (=00:30). Arrows indicate lowest light intensity before the onset of constant added light 50 lux (**A**) and 4 lux (**B**). Bars below the x-axes depict the lighting scheme (**A,B**) as lights on (white), lights off (black) or extra-light in the treatment only (grey), where upper bar = control lighting scheme and the lower bar = extra-light. The line below the lighting schemes depicts the phases of activity as analyzed in the activity experiment. Light intensity for this experiment was inadvertently reduced in the extra-light treatment due to the failure of one LED strip at the beginning of the activity experiment (**B**).

### 2.2. Expression of Circadian Clock Genes

Gene expression experiments were carried out on individuals that were reared in the control (L:D 16h:8h) environment. We were interested in gene-expression changes as an immediate response to ALAN while avoiding potential long-term effects. We therefore sampled a subset of adult mosquitoes (5–10 days old) from the control population and sorted them by sex prior to gene expression experiments. Two transparent sample containers (50 mL), one with 10 females and one with 10 males, were prepared for each treatment and sampling time point (Figure 2A) and for each of three exposure times (1, 2, 4 days) for a total of 120 containers ( 600 individuals per treatment). Each container was covered with mesh at one end to allow mosquitoes to feed on cotton pads soaked with 10% sucrose solution. Containers were placed in continual darkness for 48 hours in order to set the individuals to their endogenous clock (i.e., not entrained with environmental light regimes). After 48 h, each container of 10 individuals was placed into either the control or extra-light treatment where they were exposed to the light regime for approximately 1, 2, or 4 days (i.e., length of exposure to the light regime prior to the first sampling time point was 17.5 h, 41.5 h, or 89.5 h). Individuals were not fed for 12 h prior to sampling to minimize variance in gene expression related to digestion. One sample container per treatment and sex was removed at each of 10 time points (Figure 2A) for each length of exposure (1, 2, or 4 days). Sample containers were snap-frozen in liquid nitrogen to minimize handling of mosquitoes, and stored at −80 °C until further analysis.

From each sample container, the heads from nine individuals (randomly chosen from the 10 sampled individuals, 540 in total for each treatment) were removed on ice. Three pools of three heads each were made from each sample. We used only heads to ensure that we would capture gene expression of the central circadian clock genes because expression of clock genes differs between tissues, which could have led to ambiguous results [36,37,56,57]. Total RNA was extracted using TRIzol® Reagent (Ambion®, Invitrogen, Carlsbad, USA). Due to the small amount of starting material, the manufacturer's protocol was adjusted as follows: heads were disrupted and homogenized in 500 µL TRIzol Reagent on ice with an ULTRA-TURRAX® disperser (IKA®, Staufen, Germany) for 20 seconds (s) Chloroform (100 µL) was added, the sample was thoroughly mixed (15 s.), incubated at room temperature (10 min; m), and centrifuged (15 m., 12,000× $g$ at 4 °C). A volume of 200 µL from the aqueous phase was transferred to a new reaction tube, precipitated with 200 µL of 100% isopropanol and incubated (10 m. at room temperature, followed by incubation overnight at −20 °C). After centrifugation (12 m., 12,000× $g$, 4 °C) the supernatant was discarded and the pellet was washed with 200 µL of ice-cold 75% ethanol followed by another centrifugation step (6 m.). The supernatant was discarded and the RNA pellet was re-suspended in 20 µL RNase-free water (Carl Roth GmbH und Co. KG, Karlsruhe, Germany) and RNA content was quantified using a NanoDrop 1000 (peqlab-Biotechnology GmbH, Erlangen, Germany).

We examined changes in expression of five genes of the circadian clock (*period, timeless, chryptochrome-2, cycle, Clock*) relative to the housekeeping gene *ribosomal protein 49* (*rp49*) to test for the effect of treatment, sex, length of exposure, and sampling time. cDNA synthesis was carried out using AffinityScript Multiple Temperature Reverse Transcriptase (Agilent Technologies, Waldbronn, Germany) following the manufacturer's protocol with 14 ng cDNA/µL in a final volume of 12.7 µL. A negative control (RNase-free water instead of RNA) was included with every reverse transcription. We used a primer mix consisting of 0.25 µL oligo-dT 15 (100 pmol/µL), 0.25 µL oligo-dT 20 (100 pmol/µL) and 1 µL of random hexamer primer (100 pmol/µL) per sample. All primer sequences used in qRT-PCR were obtained from Gentile et al., [35,36]. qPCR reactions used Brilliant III Ultra Fast SYBR® Green QPCR Master Mix (Agilent Technologies) following the manufacturer's protocol and were analyzed on a Stratagene MxPro3000P or MxPro3005P (Agilent Technologies). Samples and calibrators were used for all subsequent runs in a 1:4 dilution. We added three calibrators per run, the reverse transcription negative control, and a negative control for the qPCR. All samples were analyzed twice (double determination) with the following conditions: initial denaturation at 95 °C for three minutes, then 40 cycles of 95 °C (17 s.), annealing at 60 °C (25 s.) and elongation at 72 °C (25 s.). In an additional

cycle, a melting curve was established in three steps: 95 °C (40 s.), 55 °C (30 s.) and 95 °C (30 s.; run settings: Supplementary Materials S2). Primer efficiencies were determined prior to the analyses (S2) and ranged from 90–100% except for the primers for *Clock* (82%). Melting curve analysis indicated no formation of unspecific products or primer dimers. We therefore concluded that our data were of usable quality. We used ΔCT values to first evaluate the constitutive control *rp49*, where ΔCT is the relative expression corrected for inter-run differences. Changes in expression were then determined using the ΔΔCT method [58], which quantifies the relative expression of the target genes with respect to individual *rp49* baselines.

We used two approaches to analyze the data. First, Wilcoxon sign rank tests (WSR) were used to detect significant differences in gene expression between treatments, pooling all sampling times for each sex as a simple comparison of overall changes (see Figure 3). We then used generalized linear models (GLMs) to analyze changes in gene expression as a function of treatment, sex, length of exposure, and sampling time, as well as the interactions of treatment x sex, treatment x length of exposure, and treatment x sampling time. GLM analyses were performed for each gene separately in R 3.2.0 [59]. All models were run using a Gamma distribution with log link function with the significance level set to 0.05. The optimal link function and best-fit model were determined using Akaike Information Criteria (AIC) and lowest residual deviance. To compare alternative models, we used the dropterm function in the mass package for R [60] by performing single deletion tests starting with the complete model. Within each GLM, extra-light was compared to control, males to females, 2- and 4-day exposures were compared to 1-day, and all time points were compared to time point ZT 20.5 (=00:30). When interaction terms were significant in the final model, we ran *post-hoc* comparisons (Supplementary Materials S3) using the testInteractions function in the phia package for R [61].

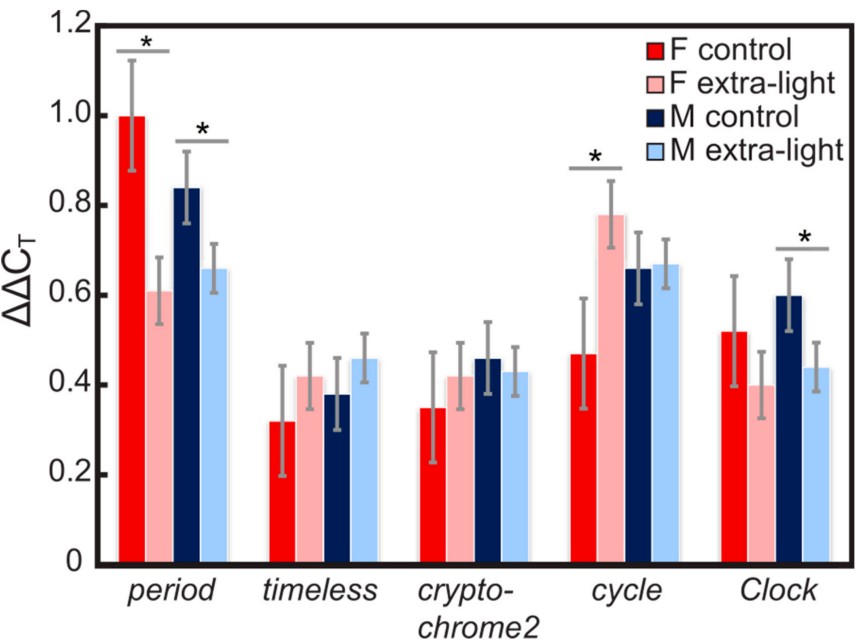

**Figure 3.** Median relative gene expression (ΔΔC$_T$) for each gene. Bars represent females (F, red) and males (M, blue) separately for each light condition (dark bars =control; light bars = extra-light; *n* = 30 for each group). Asterisks denote significant differences in gene expression between treatments calculated using WSR and pooling all sampling time points. Error bars represent the standard error.

*2.3. Diel Activity*

Individual pupae from each treatment population (control, extra-light) were transferred to separate mesh-covered containers for emergence. This was to ensure that adults were unmated and therefore that activity patterns reflected the search for food and mates but not oviposition sites. Adults were

collected within 3–4 days after emergence, and randomly selected individuals (control $n = 26$; extra-light $n = 23$) were transferred to a locomotor activity monitor (LAM, TriKintecs Inc., Waltham, USA). The LAM consists of individually monitored tubes (12.5 × 2.5 cm) and records the number of times any one of nine infrared beams is crossed within two-minute time intervals. Data acquisition began ≈7 h after individuals were placed in the tubes. Each experiment was run for five days of light:dark cycles (hereafter "LD") using either control or extra-light conditions (Figure 2B. Following the 5 days of LD, all individuals were subjected to 5 days of complete darkness (i.e., endogenous timing of activity; hereafter "DD"). Activity was not recorded for the first 24 h after individuals were moved to DD to allow for acclimation. Three individuals in each treatment died during the course of the experiment. Individuals were considered dead when there was no movement recorded for at least 12 h, in which case only data up to 1 h after the last recorded movement were used. Conditions were identical in the two treatments, except that the light intensity in the extra-light treatment was inadvertently reduced due to failure of one LED strip at the beginning of the diel activity (LD) experiment (Figure 2B; control peak = 633 lux; extra-light peak = 368 lux). This absolute change in mid-day light intensity could potentially contribute to differences in activity patterns, preventing any observation of differences caused by ALAN because the extra-light treatment was darker than the control and overall activity may be affected by daytime light intensity. However, the general patterns (onset of activity peak, active and resting phases and end of activity) were comparable in both treatments (see Results), suggesting that our data still allows us to draw conclusions about the effect of ALAN on activity.

Activity levels varied among individuals, leading to differences in the number of counts. In order to remove this bias from our estimates of activity, count data were translated into a binary (1/0 = active/inactive) matrix. The data were summed across individuals for each 2-minute time-interval (i.e., the LAM measurement interval) and divided by the number of individuals to account for the fact that three individuals died in each treatment. We used Mann-Whitney-U tests (MWU, $n_{tests} = 7200$) to compare the two treatments using two-tailed $p$-values. LD and DD stages were analyzed separately. *Culex pipiens* activity follows daily rhythms, with the main activity starting after sunset and lasting for 2–3 h [31]. The day was divided into five activity phases for further analyses: dawn (04:00–09:00 h, total 5 h), day (09:00–15:00 h, total 6 h), dusk (15:00–20:00, total 5 h), trial (20:00–23:00, total 3 h), and night (23:00–04:00 h, total 5 h, equivalent to a general pattern of 15.5 h:3.5 h:5 h L:dim:D; Figure 2B). Differences among these phases were tested using Kruskal-Wallis tests (KW, $n_{tests} = 3600$), followed by pairwise MWUs when significant. Treatment effects within each phase and sex-specific differences within treatment and phase were compared using MWU.

## 2.4. Fecundity

From each treatment population (control, extra-light; Figure 2A), a subset of pupae was transferred to a new cage where they were allowed to emerge and reproduce. The density of adults in these cages was monitored so that numbers of mosquitoes were equal in both light regimes. Dead individuals were counted and sex was determined (ratio males:females control: 1.03:1; extra-light: 1.19:1). Throughout the study (205 days) the density never exceeded 210 individuals per cage. Counts of egg rafts and eggs per raft were recorded daily. For the statistical analysis, days where no rafts were recorded were excluded. We tested for differences in the counts with a $t$-test (eggs per raft) or MWU (number of rafts). Spearman rank correlation was used to examine relationships between date and counts of rafts and eggs per raft (expressed as median number of eggs). In all tests, two-tailed $p$-values were used. We measured egg diameter in a subset of rafts during three time periods in the course of the experiment: March (i.e., 11 months after the colony was established), August, and October 2013. Diameter was measured using a microscope-attached camera (Nikon SMZ1500 and Nikon Digital Sight DS-Fi1, magnification: 100x) and the line-measuring function implemented in the NIS Elements D 3.10 software. Egg diameter measures were compared using pairwise MWU. There was a significant correlation between date and number of eggs for the control condition (see Results); therefore, we compared egg diameter in each sampling period using KW.

## 3. Results

### 3.1. Expression of Circadian Clock Genes

There was no difference in expression of the constitutive control (*rp49*) between treatments (MWU, $n = 358$, $Z = -0.819$, $p = 0.413$) or among sampling time points (KW, df = 9, control $\chi^2 = 16.473$; extra-light $\chi^2 = 8.491$, both $p > 0.05$), although *rp49* expression levels differed with exposure time (1, 2, or 4 days) within each treatment (KW, control: $n = 170$, $\chi^2 = 20.646$, df = 2 $p < 0.0001$; extra-light: $n = 187$, $\chi^2 = 10.119$, df = 2 $p = 0.006$). This difference occurred only after 1 day of exposure and likely represented an initial physiological response to the light in both treatments after 48 h of constant darkness (see Methods). ΔCT values in males were higher than females in the extra-light treatment (MWU $n = 187$, $Z = -2.066$, $p = 0.039$). These individual *rp49* baseline changes were taken into account by using ΔΔCT values to calculate changes in clock-gene expression (see below). The non-parametric test for the effect of treatment within sex (WSR; data for sampling times and exposure pooled) indicated significant differences in gene expression in three clock genes (*period, cycle, Clock*; $p < 0.05$). In extra-light, *period* was down-regulated in males and females, *cycle* was up-regulated in females, and *Clock* was down-regulated in males (Figure 3).

In the GLM analyses, extra-light treatment significantly affected the expression of all five clock genes; as a factor in *period*, and through an interaction with sampling time in *timeless, cycle, cryptochrome2*, and *Clock* (Table 1); *period* expression was reduced in extra-light compared to control (Figure 4a; *period* $b = -0.66$, $t = -6.38$, $p < 0.0001$; Supplementary Materials Table S3). In *timeless*, the significant interaction of treatment x time was driven by a single time point, ZT 15.3 ($b = -1.40$, $t = 0.25$, $p = 0.01$, Supplementary Materials Table S3), where expression was higher in the control than in the extra-light treatment and was different from the preceding and subsequent time points (Figure 4b, Supplementary Materials Table S3). The expression of *cycle* initially increased steady then dramatically beginning from ZT16.5 toward the sampling time point in the extra-light treatment (Figure 4d). Simultaneously, expression varied in the control, but was low after ZT16.5, which was also reflected in the significant interaction terms in the GLM (Table 1, Supplementary Materials Table S3). *Post-hoc* testing did not confirm treatment x time interactions to be significant for *cryptochrome2* and *Clock* genes, for which residual deviance values were also lowest (Table 1; Figure 4c,e; Supplementary Materials Table S3).

**Table 1.** Summary of GLM results for each gene with the Akaike information criterion (AIC) scores for the complete model (expression ~ treatment * (sex + exposure + time)) and best model; exposure = days of exposure to the extra-light regime (1, 2, and 4 days), time = sampling time point. Residual deviance, degrees of freedom (df) and significant terms are given for the best model for each gene.

| Gene | AIC (Complete) | AIC (Best) | Residual Deviance | df | Significant Terms |
|------|------|------|------|------|------|
| *period* | 667.23 | 660.23 | 204.77 | 346 | treatment |
| *timeless* | 323.98 | 321.33 | 219.12 | 315 | sex |
| | | | | | treatment x time |
| *cryptochrome*2 | 194.63 | 193.07 | 139.58 | 315 | sex |
| | | | | | treatment x time [a] |
| *cycle* | 672.59 | 669.28 | 205.37 | 310 | sex |
| | | | | | exposure |
| | | | | | treatment x time |
| *Clock* | 222.79 | 219.80 | 137.08 | 315 | sex |
| | | | | | treatment x time [a] |

[a] *post-hoc* test n.s.

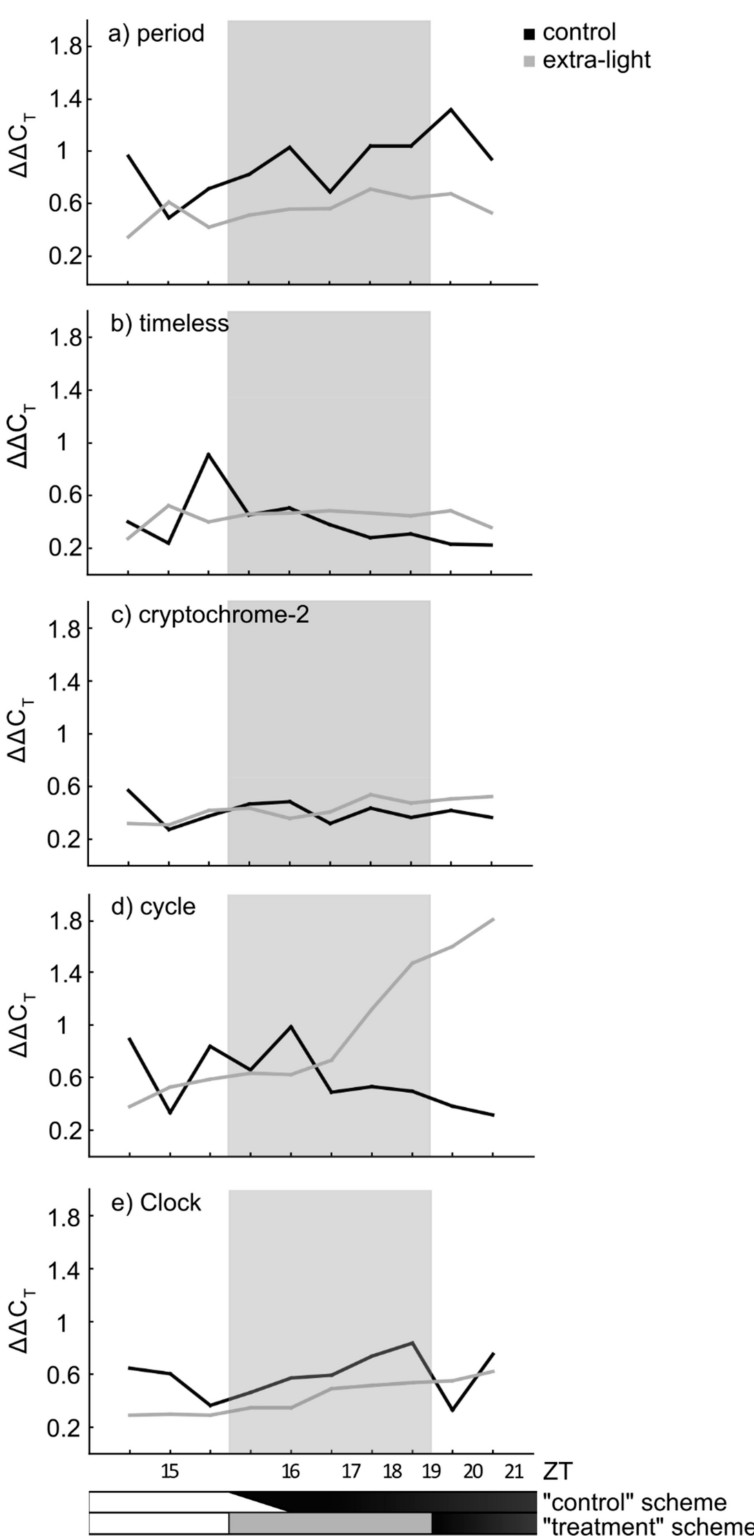

**Figure 4.** Median relative gene expression ($\Delta\Delta C_T$) of *Cx. pipiens* f. *molestus* over 10 sampling time points combining males, females and exposure times (1, 2, and 4 days). Each panel refers to one gene tested with (**a**) *period*, (**b**) *timeless*, (**c**) *cryptochrome2*, (**d**) *cycle*, (**e**) *Clock.* Shaded areas denote the time with differing light regimes (control = 0 lux, extra-light = 300 lux; Figure 2A). Black lines refer to the changes in relative expression at the sampling times in the control treatment; gray lines refer to the extra-light treatment. Sampling times are given in zeitgeber time ZT (e.g., ZT 16 is 20:00 in clock time, for reference see Figure 2). Bars below represent the light regime during sampling time.

The GLM revealed a sex-specific response for all genes except *period* (Table 1). As the expression of period was reduced in both sexes we did not detect an interaction of treatment x sex. However, the median difference in expression was larger in females (Figure 3). For all other genes, mean expression differed in males compared to females combining all available data (treatment, exposure time, sampling time) for each sex (*timeless*: $b = 0.43$, $t = 3.8$, $p = 0.0002$; *cryptochrome2*: $b = 0.23$, $t = 2.55$, $p = 0.01$; *Clock*: $b = 0.30$, $t = 3.72$, $p = 0.0002$; *cycle*: $b = 0.30$, $t = 2.78$, $p = 0.01$; Table S3); *cycle* was the only gene whose expression was also influenced by exposure days (Table 1). Mean expression was significantly lower after 2 and 4 days of exposure compared to 1 day (2 days: $b = -0.27$, $t = -2.02$, $p = 0.04$; 4 days: $b = -0.59$, $t = -4.40$, $p < 0.0001$).

### 3.2. Diel Activity Patterns

Activity was highly variable, with significant differences among phases (dawn, day, dusk, trial, night) in both stages (LD and DD) and for treatment and control individuals (extra-light: KW df $= 4$, $LD_{extra-light}$ $\chi^2 = 1860.35$, $p < 0.0001$; $DD_{extra-light}$ $\chi^2 = 1729.81$, $p < 0.0001$; control: KW df $= 4$, $LD_{control}$ $\chi^2 = 2235.08$, $p < 0.0001$; $DD_{control}$ $\chi^2 = 1503.65$, $p < 0.0001$). The only exceptions were control individuals in the DD stage, where dusk and night phases did not differ ($p = 0.164$), and dawn and trial phases did not differ ($p = 0.789$) (*post-hoc* pairwise MWU; S4).

Mosquitoes subjected to extra-light treatment during the LD stage were less active than control mosquitoes in all phases except during the day, when activity was lowest in both treatment and controls (Table 2). The results during the DD stage were similar, with those having undergone the extra-light treatment less active in all phases except the trial phase (i.e., in the early evening when ALAN had been added in the LD stage) (Table 2).

**Table 2.** Total mosquito activity (females and males combined) within the five experimental phases (dawn, day, dusk, trial, night) for the light:dark (LD) stage and constant darkness (DD) stage of the experiment. Significantly higher activity levels are indicated in bold (MWU tests; Z = test statistic corrected for ties). A full table with all test statistics can be found in the supplementary Table S5.

| Stage | Phase | Mean Activity Control | Mean Activity Extra-Light | *Z-Score* | $P_{2tailed}$ |
|---|---|---|---|---|---|
| **LD** | dawn | **0.098** | 0.053 | −3469 | 0.001 |
| | day | 0.010 | 0.008 | −1267 | 0.205 |
| | dusk | **0.105** | 0.038 | −15,030 | <0.0001 |
| | trial | **0.419** | 0.051 | −27,369 | <0.0001 |
| | night | **0.251** | 0.213 | −10,052 | <0.0001 |
| **DD** | dawn | **0.209** | 0.074 | −19,549 | <0.0001 |
| | day | **0.076** | 0.050 | −10,105 | <0.0001 |
| | dusk | **0.314** | 0.234 | −9537 | <0.0001 |
| | trial | 0.172 | **0.200** | −8695 | <0.0001 |
| | night | **0.295** | 0.114 | −30,723 | <0.0001 |

When examined separately, females were more active than males throughout the experiment except during dusk and night phases in the DD stage, when activity was similar to that of males (Figure 5; MWU; $LD_{extra-light}$, $p < 0.0001$, $LD_{control}$: $p < 0.0001$; $DD_{extra-light}$: $p < 0.0001$, $DD_{control}$: $p < 0.0001$; Table S5). During the LD stage of the experiment, treatment individuals of both sexes were much less active than controls during the trial phase, i.e., during the addition of extra light to treatment individuals (Figure 5a). At night, after the extra-light treatment ended, both females and males increased their activity, with females becoming significantly more active in treatment condition than in control (Figure 5a). During the DD stage, there was reduced diel variation compared to LD and activity was highest during the dusk phase, i.e., earlier than in the LD stage (Figure 5b). Comparing treatments, patterns were reversed from LD: extra-light individuals were less active during night phase and more active during dusk and trial phases (Figure 5).

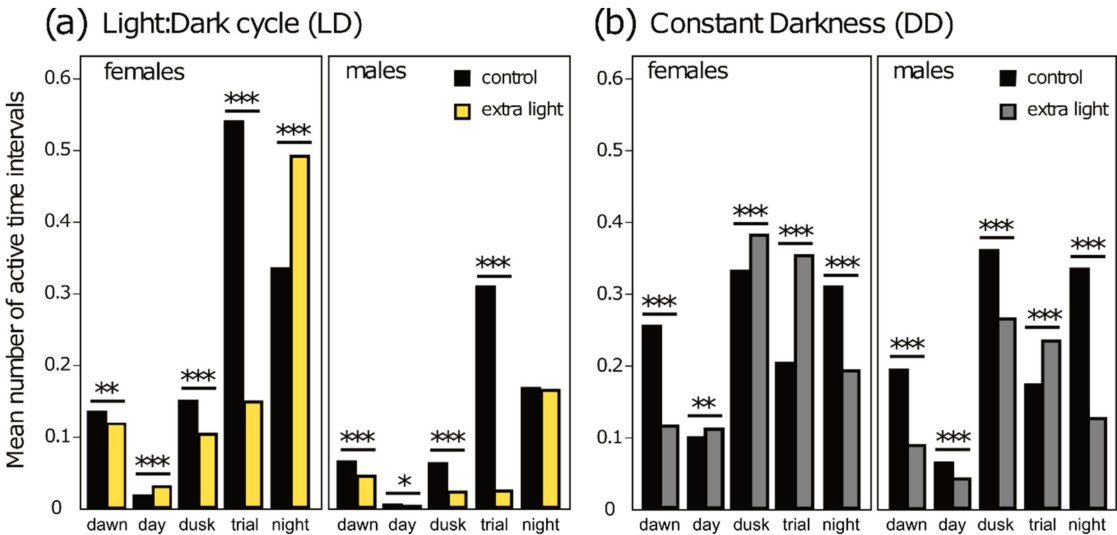

**Figure 5.** Activity of *Cx. pipiens* f. *molestus* as the mean count of active 2-mute time intervals across individuals (**a**) during light:dark cycles (LD) and (**b**) in constant darkness (DD). Daily activity was separated into 5 phases (dawn, day, dusk, trial and night), with trial representing the period during which control individuals (black bars) experienced normal darkness and extra-light individuals received constant, low-level light (yellow bars) in LD (**a**) (see also Section 2.3). Asterisks denote significant differences between treatments for females and males separately (panels). Note that error bars are not depicted because they were small due to the high number of observations and did thus not add information to the graph.

### 3.3. Fecundity

Over the duration of the study, the median number of eggs per raft was significantly higher in the control compared to extra-light conditions ($t = 3.21$, df = 92, $p = 0.002$, $n_{control} = 42$, $n_{extra-light} = 52$). The number of eggs per raft changed over time in the control (Spearman's *rho* = −0.45, $p = 0.003$) but not in the extra-light treatment (*rho* = 0.16, $p = 0.269$). The number of egg rafts produced did not differ significantly between the two treatments (MWU; $Z = −0.36$, $p = 0.718$, $n_{control} = 48$, $n_{extra-light} = 60$) and did not change over time (control: *rho* = −0.18, $p = 0.212$, extra-light: *rho* = −0.08, $p = 0.532$). Egg diameter varied over time in the control and in extra-light except between March and October (MWU; $Z = −1.154$, $p = 0.248$, $n_{March} = 88$, $n_{October} = 224$; S7), as revealed by pairwise comparisons. Because there was a significant effect of sampling period on diameter (KW; $\chi^2 = 90.97$, df = 2, $p < 0.0001$) we compared the two treatments for each period separately. Eggs in the extra-light conditions were larger in March but smaller in October compared to eggs subjected to extra-light (Table 3, Figure 6).

**Table 3.** Mean egg diameter in control and extra-light treatments measured in three periods of the study. Significantly greater diameters (KW tests) are indicated in bold.

| | | Control | | Extra-Light | | | |
| Sampling Period | *n* | Mean Egg Diameter in mm (SD) | *n* | Mean Egg Diameter in mm (SD) | *P*-Value | $\chi^2$ | df |
|---|---|---|---|---|---|---|---|
| March | 93 | 0.105 (0.015) | 88 | 0.118 (0.017) | <0.0001 | 14.29 | 1 |
| August | 110 | 0.134 (0.025) | 125 | 0.128 (0.024) | 0.47 | 3.93 | 1 |
| October | 271 | **0.125** (0.015) | 224 | 0.119 (0.014) | <0.0001 | 23.98 | 1 |

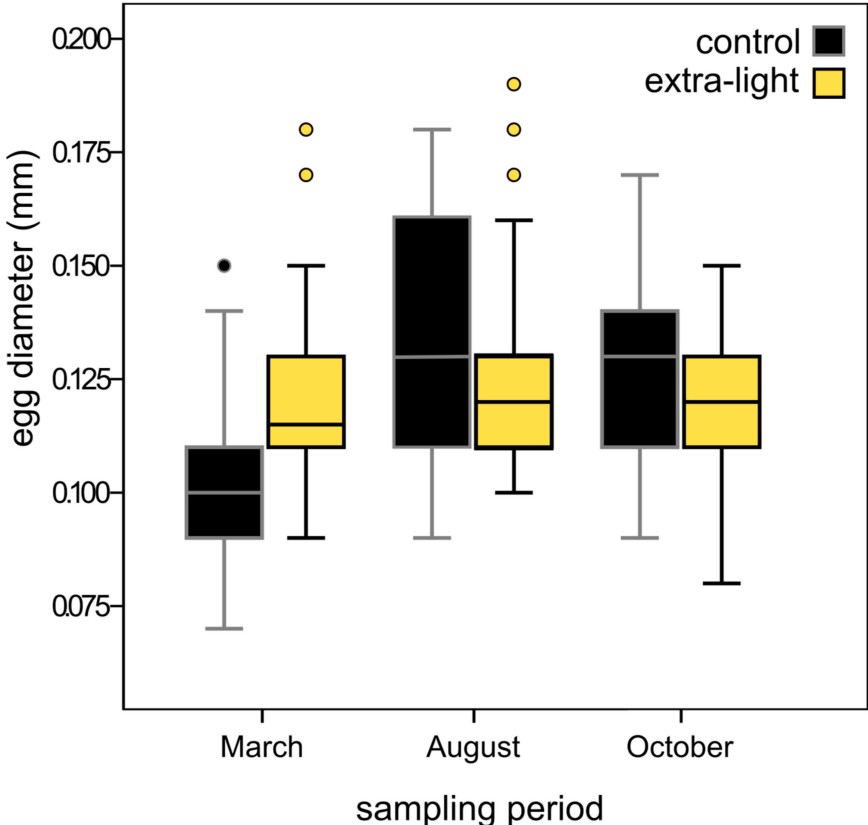

**Figure 6.** Egg diameter in "control" (black boxes) and "extra-light" (yellow boxes) treatment over the course of the study. Boxplots indicate the median egg diameter with 25% and 75% quartiles and whiskers indicate the 1.5 interquartile range, and circles signify outliers.

## 4. Discussion

We found ALAN to simultaneously influence clock-gene expression, activity, and reproductive output in a laboratory study of the mosquito *Cx. pipiens* f. *molestus*, a widespread temperate species occurring in urban and suburban areas [22,23]. In addition, alterations of gene expression and activity were sex-specific. The consequences of a sex-specific effect of ALAN may have important consequences on the population level because of its potential effect on mating and the differing roles of sexes in ecosystems [62]. Sex-specific changes in gene expression might potentially shift the timing of activity, leading to only little overlap in active phase negatively influencing mating behavior, leading to reduced population sizes. We also found a reduced number and size of eggs when ALAN was present which would be in line with this hypothesis. Future studies are needed to specifically test these links.

### 4.1. Gene Expression

Genes of the central circadian clock have been well-studied in several mosquito species, including the closely related *Cx. quinquefasciatus* [36]. Past research focused on changes in day length (i.e., changes in L:D ratios) and interactions with other factors (e.g., temperature [34]). We were interested in artificial light generated by outdoor lighting, which does not simply modify L:D ratios, but consists of an abrupt switching on of lower-intensity light source and a constant brightness for some time in a period that would otherwise be dark. An important overall finding was that the patterns we observed were consistent across days of exposure to the ALAN treatment (1, 2, or 4 days), indicating re-occurring expression patterns and consistent treatment-induced changes over time. The only exception was the *cycle* gene, where expression decreased with days of exposure in both treatments. This could indicate

age-related changes in *cycle* expression patterns, although individuals were of mixed ages (5–10 days old), or a different response of *cycle* compared to the other genes that were studied.

In all five genes examined, GLMs clearly indicated that expression was reduced in individuals exposed to ALAN. This was as a single factor in *period* and as an interaction with sampling time in *cryptochrome2*, *timeless*, *Clock* and *cycle* where significant reduction in expression occurred in a subset of time points. Interestingly, there was no clear pattern among time points that would indicate a direct and short-term response to light. Changes were also not restricted to the time of "extra-light", indicating that the response to altered light regimes may be complex and subject to the different loops of the circadian clock machinery (see Figure 1) rather than consistent short-term responses. Additionally, the temporal variation was 'flattened' in *period, timeless* and *Clock* in extra-light, indicating less variation in transcript numbers. While this was not within the scope of our study it seems a promising field for future studies on the effect of ALAN. Male and female mosquitoes exhibit many sex-specific characteristics (e.g., blood-feeding in females [63,64]) and thus ALAN may affect the sexes differently. Genome-wide studies of gene expression have reported pronounced differences between males and females in molestus [65] and in *Anopheles gambiae* [56]. Here, we found sex-specific differences in gene expression in response to ALAN in all genes except in *period.* In the case of *cycle*, males and females differed in the direction of response (up-or down-regulation of gene expression). These differences may result in changes in clock regulation.

The differences we observed in gene expression are not easily compared to published reports because the only related study was carried out on the sister species *Cx. quinquefasciatus* or distantly related *Aedes aegypti* [36]. In these earlier studies, individuals were sampled every two hours over a 24-h period in 12:12 L:D, in contrast to the higher frequency sampling over a 7-h period here. This makes it difficult to estimate whether our data reflect intrinsic (i.e., species-specific) differences in clock gene expression patterns. Although closely related, *Cx. quinquefasciatus* has a tropical distribution and may thus differ in its clock-gene expression patterns because of daily and seasonal differences in light regime, temperature, and other factors relevant to the functioning of the circadian clock (reviewed in [38]). Based on previous studies in other species, we anticipated *Clock* and *cycle* to peak when the levels of *period, timeless* and *cryptochrome2* were low [36,38]. In contrast, overall temporal changes were minimal in our control treatment, with *Clock* and *period* increasing as *timeless* and *cycle* decreased. This might result in protein levels that impede CLOCK-CYCLE-complex formation possibly disrupting the feedback loop. A notable finding was a lack of linkage between *cryptochrome2* expression and activity in *Cx. pipiens* f. *molestus*. Gentile et al. [36] speculated *cryptochrome2* might be involved in controlling activity patterns based on differences in expression patterns between a diurnal (*Ae. aegypti*) and a nocturnal (*Cx. quinquefasciatus*) species. In our study of *Cx. pipiens* f. *molestus*, *cryptochrome2* was not found to cycle over time and varied little across treatments, despite pronounced differences in behavior. This may suggest that, at least in this species, activity is controlled by different genes or via post-translational regulation [66].

In order to obtain sufficient numbers of samples, we sampled individuals 5-10 days after emergence. Although this is a relatively small window of the total lifespan, individuals may have been in different gonotrophic states. Gene expression has been reported to vary in different gonotrophic states (*Anopheles gambiae*: [67]; *Aedes aegypti*: [68]). However, other processes such as digestion (of a bloodmeal) and egg formation also influence gene expression profiles [67]; it is therefore not straightforward to determine which process exerts more influence on gene expression profiles. By pooling samples, we reduced the influence of individuals' gene expression profiles on the overall outcome. On the other hand, pooling may have reduced our ability to detect differences in response to ALAN. Our data are therefore likely to underestimate its effect. In the laboratory, *Cx. pipiens* f. *molestus* females have been reported to live an average of 10 days longer than males [69], thus males and females could have been in different states of senescence, contributing to the sex-specific differences we observed. Nonetheless, the gene expression and activity experiments were completed when the individuals were within the first half of their expected average lifespan (females: mean = 42.3 days, max. = 75 days; males: mean = 32.7,

max. = 52.5 days [69]). Sex-specific stages of senescence are therefore expected to have only had marginal effects.

## 4.2. Activity

Individuals exposed to ALAN were consistently less active, except for females which were more active at night. The effect of ALAN was more pronounced in the Light:Dark (LD) experiment compared to constant darkness (DD). Females were also more active than males in nearly all phases and regardless of treatment. Sex-specific activity patterns are known from *An. gambiae* [70] where virgin females commence flight activity 5–12 min later than males. Our data suggest that there are also light-induced differences in sex-specific activity, which might add to inherent differences between male and female activity. Activity varied consistently throughout all phases in DD, suggesting that internal timekeeping was not disrupted by the extra-light treatment that both groups received prior to the experiment in constant darkness. However, the onset of activity shifted to dusk in DD (as opposed to the trial phase in LD). This might reflect earlier anticipation of darkness, and may indicate that changing light intensity, whether increasing or decreasing, was an important trigger for activity. We found very different levels of activity in all phases except for mid-day, when activity was always lowest.

Host-seeking (i.e., female) flight activity in *Cx. pipiens* was reported to be induced when light intensity dropped below 5 lux, resulting in a strong peak two to three hours after sunset with low activity until sunrise [31]. The authors also reported a shift in the timing of activity after the autumnal equinox to before sunset and suggested that this was a light-induced change in behavior because temperatures were still favorable at the time [31]. This coincides with the phases of peak activity in our study. The extra-light clearly inhibited activity although light dropped below 5 lux prior to the addition of ALAN (see Figure 2), usually a signal to commence flying. The reduced activity under the extra-light condition (i.e., prolonged resting periods) may be followed by a reduction in foraging time and thus decreased nutrient uptake. Additionally, it may lead to fewer mating encounters.

Owing to the malfunction of some lights in the extra-light treatment, the light regimes in the activity experiment differed in their daily maximum. Although the proportion of change in the extra-light treatment was the same as in the other experiments, the absolute intensity was lower throughout the day compared to control. We cannot be certain if this led to the observed differences between treatments, or how it generally influenced activity (i.e., overall activity may be affected by daytime light intensity). However, given that the absolute light intensities in our treatment group were lower than in controls, it is remarkable that the extra light produced such a strong effect (i.e., greatly reduced activity compared to dark phases within extra-light treatment). The general pattern (active in the dark vs. resting with light) in constant darkness was very similar between the treatments, suggesting that the differences in the absolute light intensity between treatments did not have a strong effect on the results.

## 4.3. Fecundity

We observed changes in the size and number of eggs per raft, indicating that ALAN may affect larval fitness (reduced egg size) and female fecundity (fewer eggs). Adult densities were maintained at comparable levels and the number of egg rafts did not differ significantly between treatments. We therefore presume that females laid eggs at a similar rate and the amount of energy used for finding oviposition sites was comparable. Smaller eggs may have been produced because adult females were smaller under extra-light, because energy allocation to egg production and provisioning was reduced, or both. The significant changes in egg diameter over time suggest that adult female size distribution was not skewed in the overall data set. The smaller egg size observed is therefore most likely a result of reduced energy allocation. Because food was available *ad libitum*, reduced feeding is one explanation for reduced resource allocation. Another possibility is that fewer females were laying eggs in the ALAN treatment. The fact that the number of rafts was equal in treatments means that fewer ovipositing females would have had to produce more rafts, which would likely mean that they allocate fewer

resources to each. However, because ovary competence is stimulated through food intake [71] and food was available, we assume all females were capable of producing eggs.

McLay et al. [51] observed a reduction in the probability of females to commence oviposition in ALAN treatments in *D. melanogaster.* However, females that did oviposit showed, similar to our study, a reduction in the number of eggs laid. Although this did not translate into differences at the juvenile phase, there was an ALAN-induced reduction in adult survival. In the mosquito *Wyeomyia smithii*, Emerson et al. [6] found that, of all fitness components measured (pupal survivorship, embryonic viability, adult longevity, fecundity as mean number of eggs per eclosing female), it was the number of eggs that was significantly reduced in a non-resonant (L:D 10:25) light environment. The mechanisms behind these changes, however, may be different in our study because *Cx. pipiens* f. *molestus* is a nocturnal species, we used different L:D ratios, added ALAN, and we measured overall population fecundity rather than individual female fecundity. Taken together, the results of these very different studies suggest that a reduction in egg production is a common response to altered light conditions during day and night cycles.

Food limitation can influence the number of eggs per clutch; e.g., adults reared under low food conditions led to daughters laying more eggs in *An. stephensi* [72]. Larval diet influenced the number of eggs in *Ochlerotatus atropalpus* [73]. In the present study, however, no developmental stage was subjected to food limitation; we can therefore exclude the possibility that larval diet influenced the outcome. Interestingly, the difference in diameter over time (205 d) (eggs were smallest in March, largest in August, and intermediate in October) occurred in both treatments but was more pronounced in the control. Our experimental set-up lacked obvious cues of seasonal change (e.g., day length, temperature). The lake water used for oviposition and rearing of larvae may have introduced a seasonal signal, although it was filtered and autoclaved to remove bacterial and other planktonic cells that might provide potential cues from the lake community. Innate seasonality might have given rise to this pattern. It is well known that there is a complex genetic basis for seasonal rhythms [74], but it is not clear how reliable seasonality will be expressed in a constant environment, and we are unaware of any studies evaluating the egg size of *Culex* over the course of a year. We cannot fully exclude that some selective processes occurred in the colony, but we believe that the fact that this pattern is detectable in both treatments and that the time since establishment of the colony is comparatively short, suggesting that the role of selective processes is negligible in producing our results.

### 4.4. Linkages among the Components of the Study

We combined multiple experimental methods to gain insight into the complex interaction of organismal responses to an important environmental cue. While our study was not designed to mechanistically link the three sets of observations, individuals subject to ALAN had reduced clock-gene expression that was more pronounced in females, with lower activity levels, and fewer eggs per raft that were smaller in size. We propose the hypothesis that individuals subjected to ALAN rested more often and fed less frequently and that these three sets of observations are therefore related. We did not measure individual fecundity, total feeding time, sugar consumption, or growth rate, so we cannot make this link with certainty. Nonetheless, it seems unlikely that the altered clock gene expression, reduced activity and decreased reproductive output are unrelated. Our observations provide insights into important processes that are influenced by ALAN. This adds to the currently sketchy body of knowledge while at the same time highlighting interesting avenues of research.

### 4.5. Considerations of the Design

The laboratory setup allowed us to control critical environmental parameters (temperature, humidity) and simulate daily light cycles. Seasonal cues were absent in order to minimize confounding factors such as shifts in peak expression of clock genes depending on day length [75]. Regularly attending the colony to monitor densities, supply food, change water, and collect egg rafts, may have introduced olfactory cues to the females. This may have lead to different results than would occur in an

environment without hosts or with hosts always present. Importantly, potential cues were introduced to both experimental groups and therefore results are comparable, albeit not directly applicable to natural populations. The climate chamber was separated into two compartments, limiting the possibilities of replication and perhaps leading to divergence between the two groups. Individuals used for the gene expression experiments were taken from a single source population (control), and largely support the differences obtained in the other two experiments. Activity in LD was similarly low during the day phase in both treatments and in constant darkness. This indicates that the groups did not differ systematically in their activity and that behaviors were triggered by the light-environment. We chose not to translocate individuals from the control group for the activity experiment because we wanted to exclude the possibility of aberrant behavior caused by a new environment and the differences of the two treatments. We therefore believe that the experimental design was adequate given the logistical constraints on any such experiment. However, additional experimentation is needed to gain a better mechanistic understanding and to establish the consistency of responses across populations.

## 5. Conclusions

The molestus-ecotype is thought to be restricted to warmer climates due to the fact that it does not undergo diapause. This is why in urban settings they are often, but not exclusively, found in below-ground structures like tunnels [22,27]. This may result in higher sensitivity to artificial light at night. On the other hand, most below-ground structures used by mosquitoes in urban areas are equipped with artificial light sources, thus populations may be adapted to it. Our study showed that artificial light at night does impact a number of relevant processes in this mosquito and that this may have negative effects on the individual and even the population as a whole. However, long-term studies are needed to clarify whether the influence we detected is indeed detrimental or whether the mosquito can avoid this, for example by adjusting behavior such that the impact is minimized.

A recent study estimating genetic divergence between pipiens and molestus revealed that processes related to the different habitats differed between ecotypes [75]. However, processes involved with body/cell maintenance (e.g., signal transducers and transcription regulators) were highly conserved between ecotypes suggesting similar effects on both [75]. This question could be addressed by directly comparing the response of the two ecotypes to ALAN. We hypothesize that, given the conserved nature of most of the examined traits, we would find similar patterns in *Cx. pipiens* f. *pipiens*. This raises important questions relevant to pest control. Changes in activity in response to ALAN, either directly or potentially as downstream effects of altered circadian regulation, might influence host-seeking behavior in mosquitoes thus altering disease transmission dynamics. It has been reported for *Aedes aegypti* that a diurnal mosquito may make use of ALAN by prolonging its host-seeking phase, thereby potentially increasing the transmission rate [76]. Nocturnal species may shift the timing of host-seeking and resting behavior, and could potentially avoid certain control measures with negative effects for nuisance control. In summary, our results highlight the importance of further investigations into the impact of ALAN on mosquitoes with respect to mosquito control and vector capacity.

**Supplementary Materials:** The following are available online at http://www.mdpi.com/2071-1050/11/22/6220/s1, Table S1: Light regime settings, Figure S1 Spectral composition of LEDs used in the experiments, S2: RT-qPCR run settings, S3: Additional methods section for GLMs and *post hoc* tests, Table S3: Detailed results of the GLM analyses, Table S4: Pairwise differences between active phases, Table S5: Full test statistics for activity analyses, Table S6: Sex-specific activity, Table S7: Pairwise comparisons of egg diameter variation over time. The datasets analyzed in the current study are available from the corresponding author upon reasonable request.

**Author Contributions:** A.-C.H. and M.T.M. conceived the study; A.-C.H. and J.L.K. carried out the research and analyzed data for gene expression; A.-C.H. carried out research and analyzed data for behavior and fecundity; A.-C.H., J.L.K., F.H. and M.T.M. wrote the manuscript. All authors read and approved the final version of the manuscript.

**Funding:** This research was partially funded by the German Federal Ministry of Education and Research (BMBF - 033L038A) project "Verlust der Nacht".

**Acknowledgments:** We thank Katrin Preuß for her tireless, invaluable help in the laboratory, Thomas Hintze for setting up the light installation and software, Georg Staaks for assisting with climate chamber logistics, and Kirsten Pohlmann, Jenny Gill, and Simon Butler for help with the statistical analysis. Additionally, we thank Norbert Becker and Katrin Huber for providing the colony-founding individuals and for help and advice with mosquito rearing, and Carla Gentile for advice on laboratory protocols. We also appreciate the comments of two anonymous reviewers that helped improve the manuscript. The publication of this article was funded by the Open Access Fund of the Leibniz Association.

**Conflicts of Interest:** The authors declare that there is no conflict of interest.

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
