# Peer review of "Artificial Light at Night Influences Clock-Gene Expression, Activity, and Fecundity in the Mosquito Culex pipiens f. molestus"

_sustainability, doi:10.3390/su11226220_

Round 1

Reviewer 1 Report

Dear authors,

You present an interesting and relevant study. There are, however, several aspects of your study which are rather unclear. Mainly the design used and the way that the results are presented.

The different lights regimes used in experiments and controls is very confusing to me. Why are there different light regimes and different additional light intensities used for the experiments and how does this influence your results. Why do you use a different daytime light regime for the control and experimental group? How does this affect your result?  Why is the timing for the addition of “extra light” different for the experiments; Fig2a there is still some light which seem more representative of a real-world situation while for the activity exp it is almost dark (4 lux) and then lights are turned on. Street lights are usually turned on before it gets dark. These aspects should be made clear in your manuscript.

I find that the results are very difficult to interpret. There seems to be a mismatch between the supplementary material and the results you give in the manuscript. Also whether these results always correspond with the statistical analyses described is unclear to me.

Below you can find some more detailed feedback. While I understand that there are quite a lot of remarks I do want to emphasize that I find it an interesting study. I hope that my feedback is useful in improving your manuscript.

Detailed feedback:

Is there a reference for the definition of 100-300 lux as a “low” light intensity?

L 73-75; It is indeed true that part-night lighting is used by several countries but usually only on a local scale. Is most light pollution not still continues throughout the night? Perhaps this statement is therefore not entirely correct.

L93 lights à light

L125 perhaps change this to something like: “We used two laboratory experiments” it otherwise seems like something is missing in the sentence.

L126 “3.5h of constant low-intensity light” I find this confusing as the distinction is often between “part-night” lighting schemes where during a part of the night light are turned off e.g. from midnight till 5 in the morning, and continues light where lights are turned on for the entire duration of the night.

L128-130 this light intensities: should be singular instead of plural? And to what intensity are you referring? You have not yet given any except for the description of “low-intensity” at L126

L133-134 From the description given here for exp 2 it is very unclear to me what you have done. You measured activity over 24h for a group of individuals exposed to the altered light regime? What is your control group? Also for exp1 it is unclear what you control measurements are.

L134-136 It is unclear to me whether this is part of the second experiment or if it is a different experiment.

L167 do you have a reference for these values or did you measure them yourself?

L 180 To what level in the cage does this correspond?

L186 So the gene expression and fecundity were studied in two different experiments?

Figure 2 The graph does not match with the bars for the light regime. The graph indicates that “dim lights are turned on at 15:30 and the bars show 1600 also the dark period does not match for the controls.

L189 This constant added light is the intensity used for the gene expression and fecundity experiments? Also Fig 2A show for control a max value of about 850 lux and for the “extra light intensity” about 900. So you have taken 30% from which value? And how can this correspond to 300 lux instead of 270 lux?

L197 It is unclear to me what the effective sample size is. This is important also considering the initial complexity of the full model given at L255-257

L208 length of exposure to the light regime

L257 two-minute?

L276 please use ≈ to indicate “almost equal to” ~ is used in mathematical equations and is not the same as ≈

L283-285 This should be made clear much earlier. Perhaps it would be good to also include this in your figure caption

L268 In this part n is used for both the number of individuals as well as the number of sample points. Which is rather confusing. For example at L302 you probably do not have 3600 individuals or a sample size of 3600 but you have multiple measurements of a number of individuals. This should be made clear. It is unclear to me how you take into account that you have repeated measurements of the same individuals.

L299-301 you are inconsistent in the way that you write down time and durations. e.g. 2-3 h but also 09:00h and 20:00. Also the 5 phases that you created do not match with the 3 phases of your lighting scheme

L340 It is unclear to me how the results from figure 3 can be reconstructed from the material given in the supplementary material. E.g. for period there is no difference between males and females but in the figure this is given and the difference between control and extra light appears to be different for males and females (even if this is insignificant). But this is not given in the supplementary material.

Also it would be good to keep the same order in which you treat the different genes for both the figure and the supplement

For Clock the figure suggests an interaction between sex and treatment. This is not given in the supplement or in Table 1.

Why do you give the “Median” and not the mean? Your supplement also suggests that your statistical results give you means and not medians.

L379—381 Decimals are not given in the figure so this information seems obsolete

L381-382 I assume you refer here to the different panels and the results of which genes you are showing? This should come earlier in the caption, please also make a sentence of it and the information does not correspond with that given in the figure itself.

L402-405 Did you test for the interaction sex x treatment? Which seems necessary for this statement.

Figure 5 please also show some form of variation around your means; e.g. the standard error as used in other figures.

Figure 6 please include in your caption an explanation of what your boxplot is showing. I assume median with quartiles and outliers? Y axis should be Egg diameter

L442-450 This part is unnecessary as it is mostly a repeat of the introduction albeit with less references. The first paragraph should ideally concisely state what you found in your own study. So the second paragraph can be used as the first one. Preferably the paragraph would be more general before you delve into the possible consequences of your findings.

L637-652 It is not very clear how this is a consideration of your design. Secondly I would suggest you end your manuscript with a conclusion or at least a stronger paragraph than the one you now give.

Author Response

Dear referee,

thank you very much for reviewing our manuscript and for putting forward constructive comments regarding our manuscript.

We have revised our manuscript paying particular attention to the description of the light regime. In the section "Considerations of the design" we detail all aspects of our experimental designs and this did or did not influence the outcome of our experiments.

There is no mismatch between the results and the supplementary. We highlighted this in the detailed comments below. There you will also find where we changed the manuscript enhance clarity about the different analysis.

Copied below are your detailed comments. We begin our response with ##

Is there a reference for the definition of 100-300 lux as a “low” light intensity?

## Thank you for pointing that out. Compared to daylight this would be low. However, to avoid confusion we no longer use the term “low” here (line 27).

L 73-75; It is indeed true that part-night lighting is used by several countries but usually only on a local scale. Is most light pollution not still continues throughout the night? Perhaps this statement is therefore not entirely correct.

## We double-checked whether the citation here is correct, which it is. The local occurrence of ALAN in Berlin, Germany, is mentioned here as well as that in some areas only part of the lighting is switched of. Another example is the town of Balaguer, Spain, where a switch-off between 23:30 and 23:42 was observed that comprises 280 lamps, which is about 9% of the 3131 lamps and resulting in a 20% decrease of skyglow under clear sky conditions (Jechow et al. 2018). Organisms that are living in the near vicinity of such lights experience this as a switch-off of almost all ALAN. We agree that light pollution is continued through the entire night in many places. However, the point that we wish to emphasize is that the scenario we tested is a real life case (line 73-75).

## Jechow, A., Ribas, S. J., Domingo, R. C., Hölker, F., Kolláth, Z., & Kyba, C. C. (2018). Tracking the dynamics of skyglow with differential photometry using a digital camera with fisheye lens. Journal of Quantitative Spectroscopy and Radiative Transfer, 209, 212-223.

L93 lights à light

## We changed this to the singular form (line 93).

L125 perhaps change this to something like: “We used two laboratory experiments” it otherwise seems like something is missing in the sentence.

## We changed the sentence such that we refer now to “three laboratory experiments” reflecting the three different sets of measurement we took and report in this manuscript (lines 126-128).

L126 “3.5h of constant low-intensity light” I find this confusing as the distinction is often between “part-night” lighting schemes where during a part of the night light are turned off e.g. from midnight till 5 in the morning, and continues light where lights are turned on for the entire duration of the night.

## Constant low-intensity light is a term that is used to compare to natural daylight and not to different forms of artificial light occurring at night. However, to avoid confusion we deleted the terms “constant” and “low” and re-phrased the sentence slightly to enhance clarity (lines 126-128).

L128-130 this light intensities: should be singular instead of plural? And to what intensity are you referring? You have not yet given any except for the description of “low-intensity” at L126

## We have recast this sentence to read “The aim was to mimic a lighting scheme typically found in urban environments whereby nocturnal individuals experience an extended dusk which could disrupt the normal cue for the onset of daily activities, e.g. foraging, during the night [31] (lines 128-1-132). Indeed we are referring to fluctuations in natural light intensities around dusk. We also added a sentence highlighting why ALAN would make a difference in this context (lines 132-133). We use broader terms in the introduction to present the general idea and specify the experimental details in the methods section (lines 126-140).

L133-134 From the description given here for exp 2 it is very unclear to me what you have done. You measured activity over 24h for a group of individuals exposed to the altered light regime? What is your control group? Also for exp1 it is unclear what you control measurements are.

## We altered the description to increase clarity. Here, we just introduce the experiments themselves. Details are given in the methods section where there is more room to describe all experiment-specific details (lines 126-140).

L134-136 It is unclear to me whether this is part of the second experiment or if it is a different experiment.

## The description of the sentence has been changed and it should become clearer now that in the second experiment, mosquito activity was measured for 5 days in a light-dark cycle followed by five days of measurement in constant darkness (lines 136-138).

L167 do you have a reference for these values or did you measure them yourself?

## These were used to exemplify values that may be find in different urban structures and those values are not derived from our own measurements. However, to avoid confusion we removed this sentences here (lines 168-172) but added theses values to the figure caption for the sake of comparison to real-life scenarios. We added the respective references to the figure caption (lines 194-196).

L 180 To what level in the cage does this correspond?

## This refers to the bottom of the cage. We added this information to the description (line 185).

L186 So the gene expression and fecundity were studied in two different experiments?

## Yes, we consider these to be two different experiments, carried out using the same lighting regime (Fig.2 A). Only the activity study has a differing light regime because during the set-up of the activity monitor the lighting changed due to technical failure of some of the LED’s.

Figure 2 The graph does not match with the bars for the light regime. The graph indicates that “dim lights are turned on at 15:30 and the bars show 1600 also the dark period does not match for the controls.

## The figure has been adjusted and the caption has now a more detailed description to make the light regime clearer (lines 191-205). We would like to highlight that this graph depicts zeitgeber time not clock time. Indeed the dim light regime starts before the control light regime is entirely dark. This is a technical issue arising from fitting a hermite spline curve through fixed voltage levels (as explained in the methods section). This was done because for this experiment we wanted a smooth, more natural, transition in light levels as opposed to switching light intensity levels without transition. While in the control regime we aimed at having the same kind of transition at “sunrise” and “sunset” it was important for the treatment light regime that from the initial transition to low lights we were able to create a switch. This is mimicking the behavior or streetlights which a fitted with sensors and are switched on once a certain ambient light level is reached.

L189 This constant added light is the intensity used for the gene expression and fecundity experiments? Also Fig 2A show for control a max value of about 850 lux and for the “extra light intensity” about 900. So you have taken 30% from which value? And how can this correspond to 300 lux instead of 270 lux?

## This is correct, and we have clarified this in the text (line 176-177) and in the legend of Fig. 2 (line 194-195). We have removed the “30%” to eliminate confusion and simply state the lux values.

L197 It is unclear to me what the effective sample size is. This is important also considering the initial complexity of the full model given at L255-257

## We prepared a total of 120 sample containers ( = 2 treatments * 2 sexes * 10 time points * 3 exposure lengths) and 600 individuals of which 540 have been used in the analysis. This has been clarified in the text (lines 215-216 and 228))

L208 length of exposure to the light regime

## This was added (line 215)

L257 two-minute?

## In L275 was a typo, it is indeed two-minute time intervals (new line 289).

L276 please use ≈ to indicate “almost equal to” ~ is used in mathematical equations and is not the same as ≈

## Changed throughout the manuscript as suggested.

L283-285 This should be made clear much earlier. Perhaps it would be good to also include this in your figure caption

## We have added the information to the legend of Fig. 2 (first referenced at line 176): “Light intensity in the extra-light treatment was inadvertently reduced due to failure of one LED strip at the beginning of the diel activity experiment (B).” was added to the figure caption (lines 203-206). Further, figure 2 has been improved and clearly highlights the different light regimes used.

L268 In this part n is used for both the number of individuals as well as the number of sample points. Which is rather confusing. For example at L302 you probably do not have 3600 individuals or a sample size of 3600 but you have multiple measurements of a number of individuals. This should be made clear. It is unclear to me how you take into account that you have repeated measurements of the same individuals.

## We use n to refer to the number of individuals (lines 286-287) and we account for multiple measurements as follows: “Activity levels varied among individuals, leading to differences in the number of counts. In order to remove this bias from our estimates of activity, count data was translated into a binary (1/0 = active/inactive) matrix. The data were summed across individuals for each 2-minute time-interval (i.e. the LAM measurement interval) and divided it by the number of individuals.” (lines 306-310). To clarify when we use n in the context of number of test done we now write ntests (lines 310 and 316).

L299-301 you are inconsistent in the way that you write down time and durations. e.g. 2-3 h but also 09:00h and 20:00. Also the 5 phases that you created do not match with the 3 phases of your lighting scheme

## We use “2-3 h” because we generally refer to the duration of mosquito activity as described in the literature on C. pipiens activity. This time span is described in relation to sunset because the time of sunset varies seasonally and geographically. In the activity experiment that is referred to here we use clock times when we refer to the time span of each phase in our experiment. We understand that this may seem inconsistent when compared to the experiment studying clock gene expression. We use zeitbgeber times, i. e. hours from lights on, as this is the standard description in the field. The conversion from zeitgeber time to clock time can be found in figure 2. We added the timing of the phases studies to figure 2 and added a description to the figure caption (lines 203-204). We have checked the manuscript to be sure that this is consistent.

## We did not attempt to match the activity phases with the lighting phases because we were interested in any overall temporal changes in activity through the 24-hour period, rather than immediate changes in activity during each lighting phase.

L340 It is unclear to me how the results from figure 3 can be reconstructed from the material given in the supplementary material. E.g. for period there is no difference between males and females but in the figure this is given and the difference between control and extra light appears to be different for males and females (even if this is insignificant). But this is not given in the supplementary material.

## The results presented in figure 3 are based on Wilcoxon sign rank test (see lines 267-269 and 349-352) for a detailed description) testing the effect of the extra-light treatment for males and females separately. In contrast, the supplementary gives all models tested using generalized linear models including the results for the post-hoc tests. The difference in the period gene is not visible in this data because both, males and females, respond in the same way (downregulation) just the magnitude of change is different. We added a sentence highlighting the fact that we analyzed the data using two different approaches (Line 267).

Also it would be good to keep the same order in which you treat the different genes for both the figure and the supplement

## The figure and the supplementary material have been adjusted accordingly.

For Clock the figure suggests an interaction between sex and treatment. This is not given in the supplement or in Table 1.

## The sex x treatment was not significant (i.e., not shown in Table 1).

Why do you give the “Median” and not the mean? Your supplement also suggests that your statistical results give you means and not medians.

## Again this is due to the fact, that the medians have been used in the WSR calculation and they have been plotted in figure 4 while the supplementary material details the GLMs.

L379—381 Decimals are not given in the figure so this information seems obsolete

## We removed this information but kept the referral to figure 2 for the definition of zeitgeber times (lines 397-399).

L381-382 I assume you refer here to the different panels and the results of which genes you are showing? This should come earlier in the caption, please also make a sentence of it and the information does not correspond with that given in the figure itself.

## We have revised the caption and figure 4 according to the suggestions (lines 392-394).

L402-405 Did you test for the interaction sex x treatment? Which seems necessary for this statement.

## Yes, we did test for this (line 271). It was not significant for any gene (Table 1).

Figure 5 please also show some form of variation around your means; e.g. the standard error as used in other figures.

## Due to the high number of observations the error bars become very small and do not add any information to the graph. In fact, adding errors bars reduces the readability of the graph. We added a note on this to the figure caption.

Figure 6 please include in your caption an explanation of what your boxplot is showing. I assume median with quartiles and outliers? Y axis should be Egg diameter

## Both the figure and caption have been updated adding information on the boxplots.

L442-450 This part is unnecessary as it is mostly a repeat of the introduction albeit with less references. The first paragraph should ideally concisely state what you found in your own study. So the second paragraph can be used as the first one. Preferably the paragraph would be more general before you delve into the possible consequences of your findings.

## We agree and follow the suggestion to use the second paragraph as the first of the discussion. We also expanded this paragraph slightly (lines 468-477).

L637-652 It is not very clear how this is a consideration of your design. Secondly I would suggest you end your manuscript with a conclusion or at least a stronger paragraph than the one you now give.

## We have included a sub-heading and revised this part of the manuscript toemphasize our conclusions based on our data (lines 665-691)

Reviewer 2 Report

This manuscript describes the impact of artificial light exposure on C. pipiens f. molestus, an important insect species with a high social impact. Some phenotypes related to the physiology and the behavior of these insects. The expression of five key clock genes was investigated resulting in a significant alteration of their expression pattern compared to the control.

The overall impact of this research could significantly improve the current knowledge in the field. Indeed a deep knowledge of the circadian behavior and the circadian-related gene expression could facilitate the management of mosquitoes populations.

The experimental setup is excellent and the methods are described thoroughly.
The statistical tests have been conducted properly.

The results are also well discussed without overstatements.

Figure 3 Color of bars in the chart should be improved since the colors of control females and males is very similar (at least in the PDF version I have downloaded) and could confuse the readers

l156 "Males and female mosquitoes ..." Could you be consistent with the singular/plural form?

l320-322 unclear, please revise this sentence

l429-430 the sample size reported differs from those reported in Table 3. Furthermore, the size of the August sample should be also reported in this sentence

Author Response

Dear referee,

thank you for the revision of our manuscript and your constructive comments.

We have revised our manuscript paying special attention to the figures.

Please find below our responses to your comments. We copied your comment and start our response with ##

Figure 3 Color of bars in the chart should be improved since the colors of control females and males is very similar (at least in the PDF version I have downloaded) and could confuse the readers

## We produced a color figure to make it easier to distinguished males from females and treatments from control.

l156 "Males and female mosquitoes ..." Could you be consistent with the singular/plural form?

## Changed to one form (now line 160).

l320-322 unclear, please revise this sentence

## We have revised this sentence to improve clarity (lines 334-337).

l429-430 the sample size reported differs from those reported in Table 3. Furthermore, the size of the August sample should be also reported in this sentence

## There was a typo in the sample sizes, thank you for bringing this to our attention. We recast this sentence to emphasize that we are reporting the variation of egg size over time in both treatments. In this context, we only report the comparison that showed no significant variation. However, as this is important information we added the results of these calculations to the supplementary material (S7). Table 3 refers to the comparison of egg diameter measurements between treatments. We emphasize this more strongly now (lines 451-454).

Round 2

Reviewer 1 Report

Dear authors,

I am satisfied with the changes made and your response to my comments. I think the extensive changes you have done have significantly improved your manuscript. Well done!